# A Theoretical and Experimental Study on the Potential Luminescent and Biological Activities of Diaminodicyanoquinodimethane Derivatives

**DOI:** 10.3390/ijms22010446

**Published:** 2021-01-05

**Authors:** Edison Rafael Jiménez, Manuel Caetano, Nelson Santiago, F. Javier Torres, Thibault Terencio, Hortensia Rodríguez

**Affiliations:** 1School of Chemical Sciences and Engineering, Yachay Tech University, Hda. San José s/n y Proyecto Yachay, Urcuquí 100119, Ecuador; edison.jimenezg@yachaytech.edu.ec (E.R.J.); mcaetano@yachaytech.edu.ec (M.C.); 2School of Biological Sciences and Engineering, Yachay Tech University, Hda. San José s/n y Proyecto Yachay, Urcuquí 100119, Ecuador; nvispo@yachaytech.edu.ec; 3Grupo de Química Computacional y Teórica (QCT-USFQ), Instituto de Simulación Computacional (ISC-USFQ), Departamento de Ingeniería Química, Universidad San Francisco de Quito, Diego de Robles y Vía Interoceánica, Quito 17-1200-841, Ecuador; jtorres@usfq.edu.ec

**Keywords:** luminescence, quinones, fluorescence, TD-DFT, quantum yield

## Abstract

Recently, several studies have demonstrated that diaminodicyanoquinone derivatives (DADQs) could present interesting fluorescence properties. Furthermore, some DADQs under the solid state are capable of showing quantum yields that can reach values of 90%. Besides, the diaminodiacyanoquinone core represents a versatile building block propense either to modification or integration into different systems to obtain and provide them unique photophysical features. Herein, we carried out a theoretical study on the fluorescence properties of three different diaminodicyanoquinodimethane systems. Therefore, time-dependent density functional theory (TD-DFT) was used to obtain the values associated with the dipole moments, oscillator strengths, and the conformational energies between the ground and the first excited states of each molecule. The results suggest that only two of the three studied systems possess significant luminescent properties. In a further stage, the theoretical insights were confirmed by means of experimental measurements, which not only retrieved the photoluminescence of the DADQs, but also suggest a preliminary and promising antibacterial activity of these systems.

## 1. Introduction

Photoluminescence is a type of luminescence that arises from photoexcitation of the emitting species, and then the emission of light from excited electronic states. It is of great importance in many scientific and technological fields, such as physics, chemistry, material science, and biology [1,2]. Therefore, it is important to keep in mind that the emitting species responsible for this phenomenon are known as luminophores. Luminophores are one atom or a group of atoms that have the capacity of emitting light with apparent spontaneity, due to an electronic transition. These systems are the basis of the development of luminescent materials, which in turn have potential applications in a number of new technologies [3]. Thus, a great deal of attention has been paid to thoroughly understand the properties of luminescent materials as well as the underlying light-emitting processes. Indeed, the work conducted on the discovery and development of green fluorescent proteins (GFP) was awarded with the Nobel Prize in 2008 [4].

Although, luminophores should be, in principle, capable of emitting electromagnetic radiation independently of their physical state, most of the practical applications require luminophores to be synthesized as solid-films, aggregates, or crystals [5,6]. For example, solid-state organic light-emitting diodes (OLEDs) and organic field-effect transistors (OFETs) [3,7] are considered essential components in optoelectronic or sensing applications for ecosystem monitoring and biomedical research [8,9,10].

In these regards, it is important to point out that highly conjugated molecules are the most common way to obtain photoluminescence in organic molecules. Therefore, highly conjugated species and even more aromaticity are imperative characteristics when it is desired to synthesize luminescent molecules. Additionally, it is important to mention that the vicinity of luminophores is very important, mostly at the condensed phases or at high concentrations, because in most of the cases a quenching effect is known as aggregation-caused quenching or simply ACQ is produced [11,12]. The latter effect represents an enormous limiting factor for the performance of luminescent materials, and it can be attributed to the removal of excited states caused by the strong π−π stacking interaction involving the aromatic groups of the luminophores. In order to overcome the aforementioned limitation, Luo and co-workers proposed, in 2001, a new kind of organic-luminogenic materials, which upon aggregation, increases its radiative emission, rather than decreasing it [13]. This pioneering study was followed by subsequent works where a series of silole derivatives, presenting the so-defined aggregation-induced emission (AIE), were reported [6,14]. As stated in the latter, a thorough understanding of the underlying mechanism of the AIE effect is fundamental for the conversion of materials with poor luminescent potential into building blocks of high-tech optical innovations [15,16,17]. 

During the last years, significant progress has been made in this matter, resulting in the synthesis of a great variety of molecules like hexaphenylsilole [5], cruciforms [18], diphenylbutadienes [19], tetraphenylethenes [20] and diaminodicyanoquinodimethanes [16], which showed the aggregation-induced emission phenomena. This has not only brought new insights into the light-emission processes but has widened the possibilities for the development of new applications. However, open questions are yet present in the field, and research efforts are still needed. In these regards, computational approaches represent interesting tools for the study of the fluorescence capabilities of different already designed systems. In particular, these approaches allow a better understanding of the AIEgens photoluminescence mechanism to be achieved, making possible, at the same time, to assess the molecular characteristics (i.e., geometries, conformational changes and their oscillator and dipole moments) controlling the luminescent potential of a given system [21,22,23,24].

Simultaneously, quinone derivatives are well known for their biological characteristics and are implicated in a large variety of biochemical processes [25,26]. Various quinines have presented different types of biological and pharmaceutical properties, including enzyme inhibition, antibacterial, anticancer, and antifungal activities [27,28,29]. Furthermore, 5 and 6 membered heterocyclic ring quinones bearing nitrogen and sulfur atoms, have demonstrated important antibacterial and antifungal activities [30,31,32,33]. The presence of these nitrogen and sulfur atoms helps to the desired highly conjugated characters in the DADQs and introduces the possibility to obtain interesting biological properties.

The aim of the present work is to demonstrate that theoretical calculations provide truthful information on how different substitutions for benzene functionalized DADQs can enhance or quench the luminescent properties in highly conjugated organic molecules. Once the theoretical predictions were verified, tests on the antibacterial activity of the DADQs derivatives was carried out, showing its potential as growth inhibitors for Escherichia coli DH5 strains and the importance that exert structural changes when we talk about inhibitory capacity. The *E. coli* DH5 alpha strain has been widely used in assays for anti-microbial biological activity for its secure handling and optimal growth conditions in assays for inhibition of growth. Other authors have carried out similar anti-microbial activity experiments on derivatives analogous to ours [25,26].

## 2. Results

### 2.1. Theoretical Investigation

On previous investigations, theoretical calculations have revealed that benzene functionalized DADQs presents high quantum yields (QY) because their molecular structures remain very similar between the ground and the excited states. [21,22] Therefore, a structural computational research of three molecules, 2-(4-(imidazolidin-2-yl)phenyl)malonitrile **1**; 2-(4-(1-(4-nitrophenyl)imidazolidin-2-yl)phenyl)malononitrile **2** and 2-(4-(1-(2-nitrophenyl)imidazolidin-2-yl)phenyl)malononitrile **3** (Figure 1) were analyzed to obtain insights into the possible reasons for fluorescence quenching or enhancement produced in these molecules. Compound **1** will be used as a reference, due to all the research already performed in this molecule. Additionally, compounds **2** and **3** were used as new benzene functionalized approaches to understand the changes that could be produced by a small difference in conformation. 

We performed time-dependent density functional theory calculations (TD-DFT) at the CAM-B3LYP level, including a Grimme’s type empirical dispersion correction (see Materials and Methods) and using acetonitrile as implicit solvent model. In addition to the later, wB97XD was also conducted in order to measure the effect of the functional group on the transitions. 

For a molecule to have high quantum yields (QY), the difference between its ground state equilibrium geometry and its exciting counterpart must be minimal. Notably, the ground (S_0_) and excited (S_1_) state structures of compounds **1** and **2** were found to be very similar, whereas for compound **3** the changes were much more noticeable (Figure 2). Furthermore, compound **3** in its excited state exhibits curvature in the quinone moiety bearing two cyano groups (Figure 2f. red structure). This point needs to be highlighted because structural features such as dihedral angles [34], [35] or molecular stacking [36,37,38] can drastically change light absorption-emission properties of calculated molecules. Thus, among all the different possible conformations, the predictive power of the optimized structure in regards to experiments is particularly important. 

Planarity is always a reflection of electron delocalization between different atoms: in this case, this loss of planarity reflects a loss of aromaticity in the molecule. In order to quantify this loss of aromaticity in compound **3**, we computed the Nucleus Independent Chemical Shift (NICS) indexes to estimate the aromaticity of the quinone part in CAM-B3LYP calculations. (Table 1). The indexes were computed both at the center of the quinone ring (NICS(0)) and at a distance of 1Å (NICS(1)) from the center to estimate the effect mainly produced by the pi system. 

Different types of NCIS indexes can be calculated; nevertheless, the out-of-plane NICSzz descriptor was shown to be one of the most reliable [39]. According to this descriptor, there is an increase of aromaticity upon excitation from S0 to S1 in all three compounds. This observation apparently contradicts the loss of aromaticity in compound **3** upon excitation, while the curvature of the quinone ring is observed in this molecule. NCIS indexes are a great tool to estimate aromaticity but are known to suffer from different shortcomings. In this particular case, the structural change observed is totally non-local, while NCIS indexes are evaluated at specific points. In fact, through close inspection of the indexes arises an interesting observation: there are increments of NICS(1)_zz_ by 4.4 and 3.6 for compounds **2** and **3**, respectively. While NCIS indexes cannot grasp this loss of aromaticity, it allows to conclude on a differential effect of excitation on compounds **2** and **3** regarding aromaticity. 

Since the only difference between compounds **2** and **3** is the position of the nitro group (para and ortho, respectively), the cause of the curvature encountered for the excited state of the compound **3** probably lies in the presence of the nitro group. Nevertheless, there is not only a change in the position of the nitro group between **2** and **3**, but also in its orientation. In compound **2**, the nitro group is in the same plane as the cycle, whereas a dihedral angle of *ca*. 30° is found in compound **3**. The implications of this on the electronic structure are detailed in Appendix A through the presentation of charge distribution.

Additionally, it is essential to check the oscillator strengths and dipole moments associated with each molecule due to its importance for luminescent characteristics. The dipole moment is the measurement of the polarity of a molecule, and it arises from the differences in electronegativity within the atoms forming that molecule. Moreover, the greater the difference between electronegativities, the stronger the dipole moment will be. For the DADQ derivatives, it is especially interesting to study this feature, due to its donor-acceptor nature [40,41]. Furthermore, when the electrons in these molecules are excited, they are induced to move from one orbital to another, producing a change in the dipole moment. Depending on how these dipole changes interact with the solvation shell of the system, we will expect to see a negative or a positive solvatochromic effect in the molecules [42]. Correspondingly, oscillator strength is a dimensionless quantity that expresses the probability of an electronic transition to occur and describes the strength of atomic and molecular optical transitions [43,44]. Generally, it is directly related to the intensity of emission, and the greater the oscillator strengths, the more feasible the electronic transition is [45]. In these regards, the calculated oscillator strengths and dipole moments for each of the studied compounds would provide a good idea on which of them will present luminescence. In contrast, geometry and aromaticity calculations would inform which molecules could show AIE effect and which do not (Table 2). 

### 2.2. Experimental Investigation 

Additionally, to corroborate the theoretical results obtained, the synthesis and structural characterization of **1** and **2** were performed. Finally, biological activity tests were conducted on the synthesized molecules to obtain further insight into their possible future applications. 

Compounds **1** and **2** were synthesized followed a previously reported procedure. [22] Initially, 7,7,8,8-Tetracyanoquinodimethane (TCNQ) was reacted with a sacrifice pyrrolidine to favor substitution in the geminal cyano group of TCNQ. This 7-pyrrolidino-7,8,8-tricyano quinomethane (PTCNQ) reacts with the respective amines to form compounds **1** and **2** (Figure 3) with 70% and 50% yields, respectively. Characterization of compounds **1** and **2** was carried out by analytical and spectroscopic techniques (UV-VIS, FTIR, and mass spectroscopy) allowed to corroborate the derivatives structures (See Appendix A). The synthesis of **3** was not possible, probably due to the robust stability present in one of the initial reagents.

Additionally, the emission spectra of **1** and **2** were recorded in all visible range and corrected for the blank (solvent) to perform proper emission intensity integration (Figure 4). The relative quantum yield (QY) of **2** in dimethyl sulfoxide (DMSO) as the solvent system was calculated, obtaining a value of 15.64%. Accordingly, the QY of **1** in DMSO corresponds to 10.1%. [21]

### 2.3. Biological Investigation 

The results obtained for the agar diffusion technique exhibited inhibition zones for all the tested molecules. Furthermore, it is essential to remark that the solvent system by itself shows some sort of antibacterial activity, but it is not as important as the ones observed for the studied compounds. Additionally, the antibacterial activity of the tested molecules is appreciable at concentrations of 1000 μg/mL and the 100 μg/mL, while for lower concentrations, the inhibition zones are barely perceptible, which could be mostly because of the solvent system, rather than the samples itself. In every case, inoculation was executed in three repetitions. Experiments were replicated in order to confirm the gained tendencies of results and experiment reproducibility. 

The results for the OD_600_ studies showed that all the compound presents antibacterial activity. (Table 3) (Figure 5) Additionally, it has been demonstrated that *E. coli* cultures with an OD_600_ of 0.1 and incubated in an LB (Luria-Bertani) medium present a concentration of around 8 × 10^8^ cells/mL [46,47]. 

Therefore, the solvent by itself is capable of inhibiting the *E. coli* growth in values around 8.5% in comparison with the standard conditions of growth (Blank); henceforth, the other test tubes must take in account the solvent´s activity and will be directly compared with the inhibition capacity of the latter one. Consequently, compounds **1** and **2** showed a reduction of 4.7% and 4.4 %, respectively. However, **c** reaches an inhibition capacity of 7.9% and **b** reaches the highest inhibition capacity, with a value of 14.3% (Table 3). 

## 3. Discussion

Relative to the theoretical estimation of luminescence, the ground (S_0_) and excited (S_1_) states of compound **1** and **2** hardly differ from each other, fulfilling the condition of presenting a close similarity between the relaxed structures of the excited state and the ground state without internal conversion (IC) or intersystem-crossing (ISC) influencing the photoexcitation process. Hwoever, for molecule **3** the geometrical differences are quite significant, indicating an absence or very low QY for this specie. Accordingly, using CAM-B3LYP, for **1** there are oscillator strength values of 1.232 and 1.2889 for the ground and excited states, respectively, and for **2** there are values of 0.785 and 1.0087 for the S_0_ and the S_1_ states, respectively, indicating in both cases that the electronic transitions are quantum-mechanically allowed. However, for **3**, there are oscillator strength values of less than 0.01, indicating that the transitions in this compound are restrained; therefore, any kind of fluorescence in **3** can be discarded. For the dipole moment, the presence of a donor-acceptor system in our molecules will produce an electron shift from the dicyanomethane to the amine groups upon photoexcitation. These electronic transitions give rise to a reduction of the dipole moment in the excited state of around 2.5 D for **1** and **2**. In contrast, the dipole moment for **3** increases in 1.5 D. This means that compounds **1** and **2** it is expected to observe that the fluorescence spectrum will present a negative solvatochromism (blue-shift) when the polarity of the solvent is increased, while for compound **3** it is expected a positive solvatochromism with an increase in solvents polarity. The exact same conclusions can be derived using wB97XD functional with only a slight increase in the oscillator strength for all compounds calculated. As expected, compounds **1** and **2** showed not only luminescent activities but also similar QY in the experimental tests, demonstrating the validity of the theoretical approach. 

Related to the Synthesis, **1** and **2** were obtained by similar procedures previously described with minor modifications (see Supporting Information). For preparation of **3**, there could be several reasons for the no reaction between N-(2-nitrophenyl)ethylenediamine and 7-pirrolidino-7,8,8-tricyanoquinodimethane. One possibility is the formation of an intramolecular six-member ring stabilized by a hydrogen bond of the type (N–H···O) between oxygen (O) in the nitro group and the hydrogen (H) of the secondary amine present in the molecule. Moreover, through geometrical optimizations it has been found that the six-member ring formation is a viable hypothesis, due to the distance of 1.82Å and angle of 110° that correspond to the expected values for a hydrogen bond and a six-member ring. This will highly reduce the reactivity of N-(2-nitrophenyl)ethylenediamine avoiding the formation of **3** (See Supporting Information). 

The relative measured quantum yields in DMSO correspond to 10.1% and 15.64% for **1** and **2**, respectively. These results were obtained from measurements of compounds **1** and **2** under solution. Generally, the fluorescence quantum yields (QY) in a solution for these compounds are below 0.5% due to non-radiative relaxations of the excited states [22,48]. Therefore, the obtained results are very encouraging, and it will be desired to study the fluorescence QY of studied compounds in solid-state. Additionally, the different densities between the ground state and excited state illustrate a shift in electron density between the dicyanomethane moiety and the diamine changeable groups upon photoexcitation. These changes lead to a reduction in the dipole moments of the molecules in the excited state, producing a negative solvatochromic effect. These quantum yield measurements show that N-substitutions could exert essential changes in the quantum yield values of DADQ’s systems without significantly modifying the positions of the emission and absorption peaks. 

On the other hand, the biological activities tests showed that **b** reduces the bacteria growth by 14%, **c** with a 7.9%, while compound **1** and **2** reduce it in 4.7% and 4.4%, respectively. Taking into account the chemical structures for each of the studied compounds and their respective inhibition capacity, it is possible that the cyano, the primary and secondary amine groups played an essential role in the antibacterial activity in each of the molecules. Finally, comparing the antibacterial activities of **1** and **2**, it seems that the presence of the nitro-benzene moiety does not contribute to biological activity. However, it is necessary to perform more investigations to obtain further insight into the possible conformational changes that could affect the antibacterial activity of the DADQs derivative.

It will be of great interest to perform more research on the AIE characteristics of **2**, which recorded quantum yields with different conditions, such as changing temperature, solvent’s viscosity, level of aggregation, or different concentrations. Additionally, for further insight into the relationship between the antimicrobial activity and the molecular structure of each compound, it would be necessary to perform the same proves with TCNQ and with 7,7-dipyrrolidino-8,8-tricyanoquinodimethane to understand the role of cyano and pyrrolidine groups for the antimicrobial activity. 

## 4. Materials and Methods 

All geometry optimizations were initially performed using a force field approach with Avogadro [49] and later with the Gaussian 16 (revision A.03) suit of programs [50]. The ground and excited state optimizations were carried out at the CAM-B3LYP/def2-TZVP and wB87XD level of density functional theory (DFT) [22,51]. The interactions of dispersion were taken into account in CAM-B3LYP calculations through Grimme’s model in the form of D3 with Becke Johnson damping. Additionally, calculations with the implicit solvent model (SCRF) were incorporated with acetonitrile as the solvent medium. Acetonitrile was used because is not a protic solvent, that could not induce fluorescence quenching by hydrogen bonding or a too viscous solvent, that would enhances the fluorescence due to its viscosity [52].

CAM-B3LYP was used for this type of calculations because it has shown an accurate performance for excited states [21,53,54,55] and species displaying a strong charge-transfer character within DFT framework. However, the functional wB97XD was also used for sake of comparison. In all cases as specified in the methodology section, the absolute minima of each structure were proved by showing no negative (imaginary) frequencies in the frequency calculations.

Synthesis procedures of **1** and **2** were carried out followed a reported procedure for analogous compounds [21]. For individual reactions yield, structures and their characterizations see supporting information.

Quantum yield measurements (QY) were performed using the comparative method of Williams et al. [56]. Hence, for this approach, **1** was used as the standard sample and **2** as the test sample. Both were diluted in DMSO at a concentration of 200 μg/mL and placed in scrupulously clean 1 cm fluorescence cuvettes. The emission spectra of both reference and sample were recorded in full and corrected for the blank (solvent only) to perform proper emission intensity integration. The configuration used for data collection was a 180° excitation-fluorescence detection geometry, with a slight deflection of the optical fiber from the excitation source to avoid any interference signal from the source. Furthermore, the absorption (optical density) of both samples was recorded under the same conditions as the emission spectrum. Finally, the relative quantum yield (QY) of the sample in the solvent used (DMSO) was calculated according to the equation (1):(1)$$\Phi_{x}=\Phi_{st}\frac{Int_{x}}{Int_{st}}\left(\frac{1-10^{A_{st}}}{1-10^{A_{x}}}\right)\frac{n_{x}^{2}}{n_{st}^{2}}$$

Antibacterial activity was tested for compounds **1**, **2**, PTCNQ and N-(4-nitrophenyl)ethylenediamine) the latter two referred to as **b** and **c**, respectively. In order to estimate the antibiotic efficacy of the synthesized molecules, two techniques were used: the agar diffusion technique and the optical density (OD_600_) measurements.

Optical density measurements were performed on a Thermo-scientific Nanodrop UV-Vis spectrophotometer. Absorbance measurements were taken every 30 min at 600 nm wavelength and room temperature. Six different test tubes were prepared with 2.5 ml of Luria Bertani (LB) broth base, and one colony of E. coli DH5-alpha strains to obtain an initial OD_600_ of 0.02. For tube 1, nothing else was added, while for tube 2, 100 μL of the solvent system (5:5; water/DMSO) was additionally added. For tubes 3, 4, and 5, 100 μL of 1mg/ml solutions of compounds PTCNQ, 1 and 2 were added.

## 5. Conclusions

Computational calculations provide geometry conformation, dipole moments, and oscillator strengths values that establish potential luminescent capacities for compounds **1** and **2**, and denies the possibility to have this activity in compound **3**. 

Compounds **1** and **2** were successfully synthesized and characterized, while a similar procedure gave no result for the preparation of compound **3**. We proposed that the unattainability of product **3** was due to the formation of a highly stable specie in one of its initial reagents, supported by theoretical calculations. The experimentally obtained quantum yields for **1** and **2** of 10.1% and 15.6%, respectively, confirm the obtained results in the theoretical section of this research. This study particularly highlights the predictive power of theoretical calculations regarding luminescence properties. Moreover, the specific geometric features (the difference between fundamental and excited state) and aromaticity displayed in the calculations would affect the π−π stacking of these molecules, giving essential clues in the context of AIE effect study.

The biological activity test of these molecules reveals an inhibition capacity to *E. coli* strains, which could be related to the presence of cyano groups in the molecular structure of the studied molecules. In summary, synthesized diaminodicyanoquinones (DADQ´s) are valuable customizable fluorescent systems with potential photophysical and bio-luminescent applications. 

## Figures and Tables

**Figure 1 ijms-22-00446-f001:**
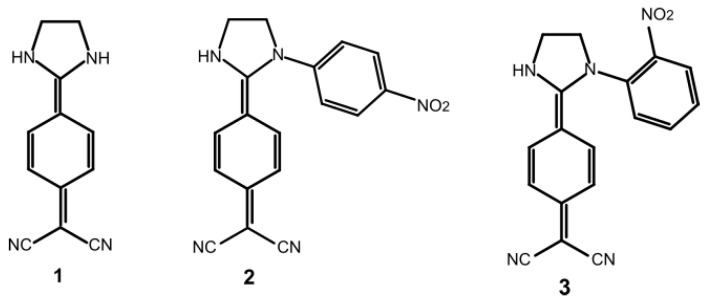
Molecular structure of the target molecules.

**Figure 2 ijms-22-00446-f002:**
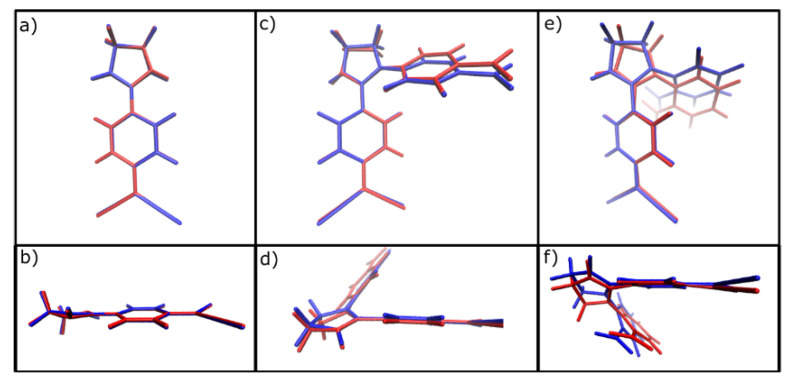
Comparison between the optimized ground state (blue) and excited state (red) structures. (**a**) frontal view of molecule 1; (**b**) lateral view of molecule 1; (**c**) frontal view of molecule 2; (**d**) lateral view of molecule 2; (**e**) frontal view of molecule 3; (**f**) lateral view of molecule 3.

**Figure 3 ijms-22-00446-f003:**
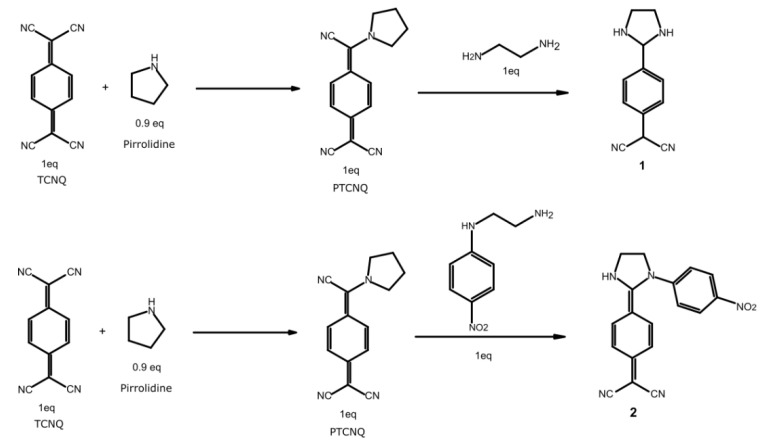
Synthetic route starting with the activation of TCNQ, to obtain molecules **1** and **2**.

**Figure 4 ijms-22-00446-f004:**
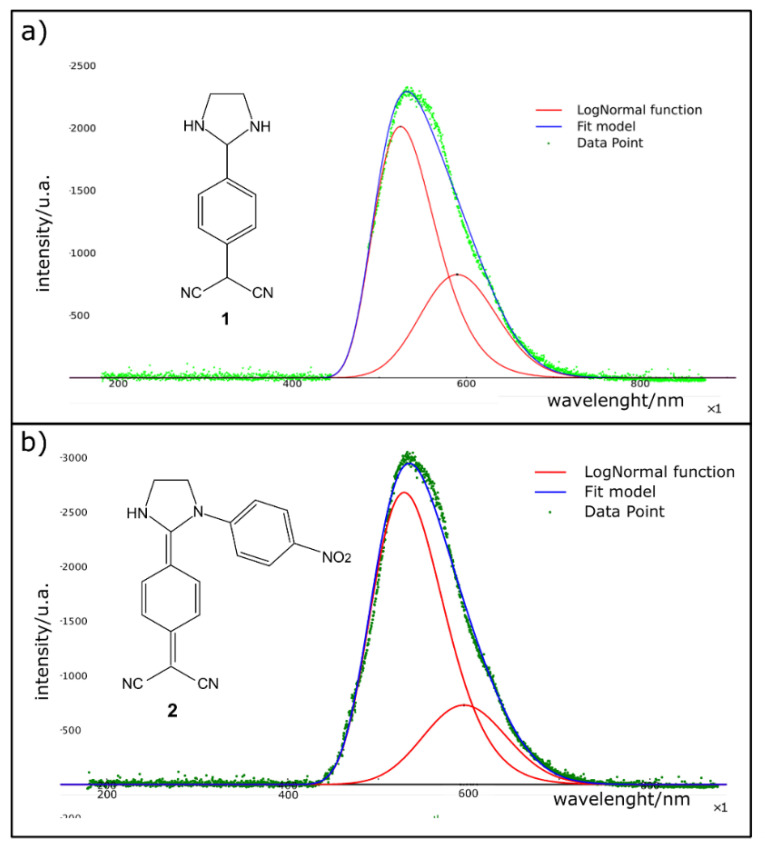
Recorded emission spectra in all the visible range using a diode laser class 3R of 405 nm. (**a**) Emission spectra and its proper deconvolution of molecule **1**; (**b**) Emission spectra and its proper deconvolution of molecule **2**.

**Figure 5 ijms-22-00446-f005:**
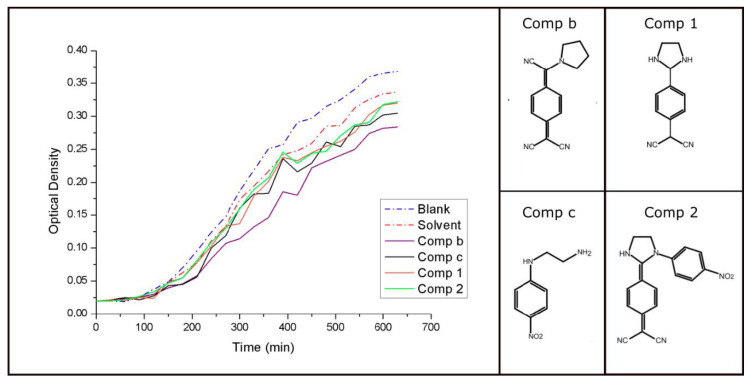
Growth curves for the OD_600_ tests performed to five different test tubes, blank (growth curve under standard conditions); Solvent (growth curve under the presence of the used solvent 5:5 water/DMSO); Comp.b (growth curve under the presence of PTCNQ) ); Comp.1 (growth curve under the presence of compound **1**); Comp.2 (growth curve under the presence of compound **2**); Comp c (growth curve under the presence of compound N-(4-nitrophenyl)ethylenediamine)).

**Table 1 ijms-22-00446-t001:** NICS indexes of aromaticity of the quinone ring in the the compounds **1**, **2** and **3**. (0) specify at the center of the ring. (1) specify it is calculated at a distance of 1Å perpendicular to the plane of the quinone ring. Iso is based on the isotropic shielding while zz is the zz-component of the shielding.

		NICS(0)_iso_	NICS(1)_iso_	NICS(0)_zz_	NICS(1)_zz_
S_0_	**1**	−4.0	−6.3	2.5	−15.8
	**2**	−3.7	−6.0	3.8	−14.1
	**3**	−4.3	−6.8	2.0	−16.2
S_1_	**1**	−5.2	−7.4	−1.9	−19.3
	**2**	−5.5	−7.6	−3.2	−18.5
	**3**	−5.7	−7.6	−3.1	−19.8

**Table 2 ijms-22-00446-t002:** Oscillator strength and dipole moments of compounds **1**, **2**, and **3** in the ground and excited-state calculated using CAM-B3LYP and wB97XD functionals. S_0_− > S_1_ transitions are reported for ground state geometries.

		CAM-B3LYP			wB97XD		
		1	2	3	1	2	3
Oscillator Strength	S_0_	1.2520	0.7850	0.0030	1.1755	0.9423	0.0161
	S_1_	1.2889	1.0087	0.0005	1.3028	1.1231	0.0005
Dipole moment (Debye)	S_0_	29.3167	31.0528	30.5951	29.7543	31.4340	29.9483
	S_1_	27.1140	28.5555	31.0556	27.1819	28.1637	30.2726
S_0_ − > S_1_ transition (eV)		3.15	3.08	3.02	3.41	3.23	3.31

**Table 3 ijms-22-00446-t003:** Final OD_600_ values, the estimated number of cells per ml, and percentages of inhibition for each growth curve.

	OD_600_	Cells/mL	% Cells Growth	% of Inhibition
Blank	0.365	2.92 × 10^9^	100	0
Solvent	0.334	2.672 × 10^9^	91.5	8.5
Comp b	0.282	2.256 × 10^9^	77.2	14.3
Comp c	0.305	2.440 × 10^9^	83.5	7.9
Comp 1	0.317	2.536 × 10^9^	86.8	4.7
Comp 2	0.318	2.544 × 10^9^	87.1	4.4

## Data Availability

All the optimised structures in ground state and excited states are available in the Appendix A.

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
