# Peer review of "A Theoretical and Experimental Study on the Potential Luminescent and Biological Activities of Diaminodicyanoquinodimethane Derivatives"

_ijms, 2021, doi:10.3390/ijms22010446_

Round 1
Reviewer 1 Report
Jiménez et al presented theoretical study on three diaminodicyanoquinone derivatives (DADQs) and subsequent experimental analysis to show that two of those systems had the potential to be employed as fluorophores due to their unique properties. Furthermore, …. Below are my suggestions.
- In the introduction, the terms ‘luminescence’ and ‘fluorescence’ and ‘luminophores’ and ‘fluorophores’ have been used interchangeably. It would be helpful for the reader to have these concepts briefly defines, especially pointing out the difference. For example. I would consider green fluorescent proteins (GFP) (line 38) to be a fluorophore.
- Introduction lacks previous studies that are similar to the one reported in this manuscript.
- Introduction, last paragraph, context should be provided for motivation to study antibacterial effect of DADQs. In its current form, it is confusing why the authors are interested in antibacterial effects? Is this something DADQs are known for, or has it been shown in other DADQs systems? Please elaborate.
- Motivate the use of particularly Escherichia coli DH5 strains for this study
- Results section 2.1. How did the authors decide to investigate the three particular molecules?
- Please define each of the parameters investigated : dipole moments, oscillator strengths and conformational energies
- For biological investigation, section 2.3, hoe many repeats/iterations were performed/
- Section 2.3, statistical tests should be performed to show the difference in OD600 growth percentage is significantly different.
Author Response
POINT-BY-POINT RESPONSE TO REVIEWER 1
All suggestions and comments have been incorporated in the second version of this paper and highlight in yellow.
Points to address:
- In the introduction, the terms' luminescence' and 'fluorescence' and 'luminophores' and 'fluorophores' have been used interchangeably. It would be helpful for the reader to have these concepts briefly defines, especially pointing out the difference. For example. I would consider green fluorescent proteins (GFP) (line 38) to be a fluorophore.
Done. An introducing paragraph (Lines 35-40, REF 1, and 2) have been included in the revised version, where the mentioned concepts have been briefly described. A brief definition of photoluminescence and what a luminophore is was added at the beginning of the introduction. The term Fluorophore is not used during the article.
2. Introduction lacks previous studies that are similar to the one reported in this manuscript.
Done. A few lines giving examples of different AIE systems discovered during the years and new references about the latest examples and previous studies related to our research were added to the introduction (See Lines 68-71).
3. Introduction, last paragraph, context should be provided for motivation to study antibacterial effect of DADQs. In its current form, it is confusing why the authors are interested in antibacterial effects? Is this something DADQs are known for, or has it been shown in other DADQs systems? Please elaborate.
Done. A new paragraph with essential references for the topic has been added. In which it is explained that most of quinone derivatives possess biological or pharmaceutical characteristics, moreover the ones containing heterocycles of 5 and 6 members are well known for presenting antibacterial and antifungal activities. That's why it will be interesting to test the biological activity of the synthesized DADQs, because these molecules contain the desired characteristics to present biological activity. Additionally, small changes were performed to the final paragraph of the introduction (See Lines 89-95).
4. Motivate the use of particularly Escherichia coli DH5 strains for this study.
The E. coli DH5 alpha strain has been widely used in assays for anti-microbial biological activity for its secure handling and optimal growth conditions in assays for inhibition of growth. Other authors have carried out similar anti-microbial activity experiments on compounds analogous to ours. (See Lines 92-95).Two new references has been also included.
5. Results section 2.1. How did the authors decide to investigate the three particular molecules?
Done. A better explanation on how we decided to study the three main molecules was added to section 2.1 (See Lines 99-101 and Lines 105-108).
6. Please define each of the parameters investigated : dipole moments, oscillator strengths and conformational energies.
Done. Definition related the dipole moment, the oscillator strength, and conformational energies, but also their relevance for the studied molecules has been included (See Lines 159-170).
7. For biological investigation, section 2.3, how many repeats/iterations were performed.
For all biological test three repeats/iterations were performed. In section 2.3, the number of inoculations repletion is performed, and the replication of the tests was included (See Lines 205-208).
8. Section 2.3, statistical tests should be performed to show the difference in OD600 growth percentage is significantly different.
All tests were performed autonomously at slightest three times. Experiments were replicated in order to confirm gained tendencies of results and ensure reproducibility. (See Lines 205-208).

Reviewer 2 Report
The present manuscript reports on emission properties of a set of diaminodicyanoquinodimethane derivatives.
Some concerns on the computational framework are reporting in the following.
In particular, it is mostly appropriate to calculate the oscillator strengths at a given excitation energy, so that for each transition corresponding to a given wavelength a value of oscillation strength is reported.
The supporting information shows some snapshots of optimized conformations as a function of the dihedral angle. I suggest to calculate the oscillator strengths for those conformations because it has recently shown that dihedral angle variations can affect the absorption properties (Pietropaolo, Nakano, Chirality, 2020 32, 661-666; Cozza et al. J. Chem Theory Comput. 2018,14:5441–5445) as well as the molecular stacking Pietropaolo et al. Nanoscale, 2017, 9:4989-4994; Brédas et al. Chem. Rev. 104, 4971; Thomas et al. Nature Commun. 2614, 2019).
Furthermore, the Materials and Methods section reads “CAM-B3LYP was used for this type of calculations because it has shown a very accurate performance for excited states and species displaying a strong charge-transfer character. “
This sentence is not correct. DFT methods in general are not suited to study charge-transfer excitations. In order to alleviate this shortcoming, specific functionals with long-range and dispersion corrections, like the WB97XD developed by Martin Head-Gordon and coworkers as well as the ones developed by Stefan Grimme and coworkers including the dispersion corrections.
Therefore, it is suggested also to use these functionals apart the long-range corrected "CAM-B3LYP" to predict the emission properties of the molecules under study.
Author Response
POINT-BY-POINT RESPONSE TO REVIEWER 2
The present manuscript reports on emission properties of a set of diaminodicyanoquinodimethane derivatives.
Some concerns on the computational framework are reporting in the following. In particular, it is mostly appropriate to calculate the oscillator strengths at a given excitation energy, so that for each transition corresponding to a given wavelength a value of oscillation strength is reported.
The supporting information shows some snapshots of optimized conformations as a function of the dihedral angle. I suggest to calculate the oscillator strengths for those conformations because it has recently shown that dihedral angle variations can affect the absorption properties (Pietropaolo, Nakano, Chirality, 2020 32, 661-666; Cozza et al. J. Chem Theory Comput. 2018,14:5441–5445) as well as the molecular stacking Pietropaolo et al. Nanoscale, 2017, 9:4989-4994; Brédas et al. Chem. Rev. 104, 4971; Thomas et al. Nature Commun. 2614, 2019).
Furthermore, the Materials and Methods section reads “CAM-B3LYP was used for this type of calculations because it has shown a very accurate performance for excited states and species displaying a strong charge-transfer character. “
This sentence is not correct. DFT methods in general are not suited to study charge-transfer excitations. In order to alleviate this shortcoming, specific functionals with long-range and dispersion corrections, like the WB97XD developed by Martin Head-Gordon and coworkers as well as the ones developed by Stefan Grimme and coworkers including the dispersion corrections.
Therefore, it is suggested also to use these functionals apart the long-range corrected "CAM-B3LYP" to predict the emission properties of the molecules under study.
Answer:
We want to thank the referee for reviewing this paper and for providing with his feedback about the theoretical part.
Indeed, the dihedral angle affect the absorption properties drastically and it would be interesting to explore it further but it would take additional time to calculate it and we think it is not the main focus of this article: we wanted to highlight the qualitative predictive power of theoretical methods relying on the optimized structure/conformation. But for being totally quantitative it would be indispensable to make different calculations with different dihedral angles, but was reported for similar complex. The most representative example was given by P. Rietsch et al. (doi: 10.1002/anie.201903204). However, all the references suggested by the referee were included into the article as an additional value (as well as well all the references mentioned in this answer).
We agree that DFT methods are not perfectly suited to study charge transfer excitations. Post-HF would definitely outperform DFT methods for this kind of calculation. But we wanted to express that within the DFT framework CAM-B3LYP performs relatively well for excitation calculation as demonstrated in different articles. (doi: 10.1002/anie.201903204; doi: 10.1021/acs.jctc.5b00105; doi: 10.1021/jp911329g; doi: 10.1021/acs.jctc.6b00966) One remarkable precision that we omitted in the manuscript is that we used Grimme’s dispersion in the form D3BJ for our calculations. We introduced this precision in the manuscript.
We also have followed your suggestion of using wB97XD to do the calculation. As it includes D2 dispersion it is an excellent complement to the article. The results obtained with this functional are very close to the ones obtained using CAM-B3LYP.

Round 2
Reviewer 1 Report
The author responses are satisfactory
Author Response
Dear colleague,
Thanks. Undoubtedly, the quality of the submitted manuscript was substantially improved with all your suggestions and corrections incorporated during the revision process.
The English of current version was again revised and the fine/minor spell checked and improved.
Best regards
H
Hortensia María Rodríguez Cabrera, PhD
Dean
School of Chemical Science and Engineering
e-mail: hmrodriguez@yachaytech.edu.ec
Tf: 0994336513
Reviewer 2 Report
The authors have improved the overall quality of the manuscript.
I suggest to include in Table 2, together with the calculated values of the oscillator strength, the values of the excitation energy for the S0-S1 transition for each compound 1-3.
Author Response
Dear colleague,
Thanks. Undoubtedly, the quality of the submitted manuscript was substantially improved with all your suggestions and corrections incorporated during the revision process.
The English of current version was again revised and the fine/minor spell checked and improved. But also the Methods and Results have been revised.
Related your comment and suggestion:
I suggest to include in Table 2, together with the calculated values of the oscillator strength, the values of the excitation energy for the S0-S1 transition for each compound 1-3.
The values for the S0-S1 transition for each compound 1-3 were included in table 2 (Highlighed in green).
Best regards
H
Hortensia Rodriguez
Dean
School of Chemical Sciences and Engineering
Yachay Tech, Ecuador